# Discovered Policy Optimisation

**Chris Lu**[*]
FLAIR, University of Oxford
christopher.lu@exeter.ox.ac.uk

**Jakub Grudzien Kuba**[* †]
BAIR, UC Berkeley
kuba@berkeley.edu

**Alistair Letcher**
aletcher.github.io
ahp.letcher@gmail.com

**Luke Metz**
Google Brain
luke.s.metz@gmail.com

**Christian Schroeder de Witt**
FLAIR, University of Oxford
cs@robots.ox.ac.uk

**Jakob Foerster**
FLAIR, University of Oxford
jakob.foerster@eng.ox.ac.uk

## Abstract

Tremendous progress has been made in reinforcement learning (RL) over the past decade. Most of these advancements came through the continual development of new algorithms, which were designed using a combination of mathematical derivations, intuitions, and experimentation. Such an approach of creating algorithms manually is limited by human understanding and ingenuity. In contrast, *meta-learning* provides a toolkit for automatic *machine learning* method optimisation, potentially addressing this flaw. However, black-box approaches which attempt to discover RL algorithms with minimal prior structure have thus far not outperformed existing hand-crafted algorithms. *Mirror Learning*, which includes RL algorithms, such as PPO, offers a potential middle-ground starting point: while every method in this framework comes with theoretical guarantees, components that differentiate them are subject to design. In this paper we explore the Mirror Learning space by meta-learning a "drift" function. We refer to the immediate result as *Learnt Policy Optimisation* (LPO). By analysing LPO we gain original insights into policy optimisation which we use to formulate a novel, closed-form RL algorithm, *Discovered Policy Optimisation* (DPO). Our experiments in *Brax* environments confirm state-of-the-art performance of LPO and DPO, as well as their transfer to unseen settings.

## 1 Introduction

Recent advancements in deep learning have allowed reinforcement learning algorithms [35, RL] to successfully tackle large-scale problems [36, 34]. As a result, great efforts have been put into designing methods that are capable of training a neural-network policies in increasingly more complex tasks [33, 31, 24, 9]. Among the most practical such algorithms are TRPO [31] and PPO [32] which are known for their performance and stability [2]. Nevertheless, although these research threads have delivered a handful of successful techniques, their design relies on concepts handcrafted by humans, rather than discovered in a scalable *learning process*. As a possible consequence, many of these methods suffer from design flaws, such as the brittleness to hyperparameter settings [31, 12], or a lack of robustness guarantees.

However, the most promising alternative approach, *algorithm discovery* via meta-reinforcement learning, thus far has been a "tough nut to crack". On the one end of the spectrum, popular "black-box" approaches in meta-RL [30, 7, 4] are unable to *generalise* to tasks outside of their meta-training distribution. On the other end of the spectrum, approaches that attempt to meta-learn algorithms

---

[*]Equal Contribution
[†]Work done while at FLAIR, University of Oxford

36th Conference on Neural Information Processing Systems (NeurIPS 2022).

starting with more structure [25, 16] typically fail to outperform existing fully handcrafted algorithms. To make matters worse, approaches at both ends of the spectrum lack theoretical guarantees.

Recently, *Mirror Learning* [18], a new theoretical framework, introduced an infinite space of provably correct algorithms, all of which share the same template. In a nutshell, a Mirror Learning algorithm is defined by four attributes, but in this work we focus on the *drift function*. A drift function guides the agent's update, usually by penalising large changes. Any Mirror Learning algorithm provably achieves monotonic improvement of the return, and converges to an optimal policy [18]. Popular RL methods such as TRPO [31] and PPO [32] are instances of this framework.

In this paper, we use meta-learning to *discover* a new state-of-the-art (SOTA) RL algorithm within the Mirror Learning space. Our algorithm thus inherits *theoretical convergence guarantees* by construction. Specifically, we parameterise a drift function with a neural network, which we then meta-train using *evolution strategies* [29, ES]. Our approach is a new middle ground between black-box optimisation and utilising prior structure. The outcome of this meta-training is a specific Mirror Learning algorithm which we name *Learnt Policy Optimisation* (LPO).

While having a neural network representation of a novel, high-performing drift function is a great first step, our next goal is to *understand* the relevant algorithmic features of this drift function. Our analysis reveals that LPO's drift discovered, for example, optimism about actions that scored low rewards in the past—a feature referred to as *rollback* [37]. Building upon these insights we propose a new, closed-form algorithm which we name —*Discovered Policy Optimisation* (DPO). We evaluate LPO and DPO in the *Brax* [8] continuous control environments and *Minatar* [41, 20] environments, where they obtain superior performance compared to PPO. Importantly, both LPO and DPO generalise to environments that were not used for training LPO. To our knowledge, DPO is the first theoretically-sound, scalable deep RL algorithm that was discovered via meta-learning.

## 2  Related Work

Over the last few years, researchers have put significant effort into designing and developing algorithmic improvements in reinforcement learning. Fujimoto et al. [9] combine DDPG policy training with estimates of pessimistic Bellman targets from a separate critic. Hsu et al. [15] stabilise the, previously unsuccessful [32], KL-penalised version of PPO and improve its robustness through novel policy design choices. Haarnoja et al. [13] introduce a mechanism that automatically adjusts the temperature parameter of the *entropy bonus* in SAC. However, none of these hand-crafted efforts succeeds in fully mitigating common RL pathologies, such as sensitivity to hyperparameter choices and lack of domain generalisation [4]. This motivates radically expanding the RL algorithm search space through automated means [27].

Popular approaches in meta-RL have shown that agents can learn to quickly adapt over a pre-specified distribution of tasks. $RL^2$ equips a learning agent with a recurrent neural network that retains state across episode boundaries to adapt the agent's behaviour to the current environment [4]. Similarly, a MAML agent meta-learns policy parameters which can adapt to a range of tasks with a few steps of gradient descent [7]. However, both $RL^2$ and MAML usually only meta-learn across narrow domains and are not expected to generalise well to truly unseen environments.

Xu et al. [40] introduce an actor-critic method that adjusts its hyperparameters online using *meta-gradients* that are updated with every few inner iterations. Similarly, STAC [42] uses implementation techniques from IMPALA [5] and auxiliary loss-guided meta-parameter tuning to further improve on this approach.

Such advances have inspired extending meta-gradient RL techniques to more ambitious objectives, including the discovery of algorithms *ab initio*. Notably, Oh et al. [25] succeeded in meta-learning an RL algorithm, LPG, that can solve simple tasks efficiently without explicitly relying on concepts such as value functions and policy gradients. Similarly, Evolved Policy Gradients [14, EPG] meta-trains a policy loss network function with Evolution Strategies [29, ES]. Although EPG surpasses PPO in average performance, it suffers from much larger variance [14] and is not expected to perform well on environments with dynamics that differ greatly from the training distribution. MetaGenRL [17], instead, meta-learns the loss function for deterministic policies which are inherently less affected by estimators' variance [33]. MetaGenRL, however, fails to improve upon DDPG [21] in terms of performance, despite building up on it. Neither EPG nor MetaGenRL have resulted in the discovery

of novel analytical RL algorithms, perhaps due to the limited interpretability of the loss functions learnt. Lastly, Co-Reyes et al. [3], Garau et al. [10] and Alet et al. [1] discover and improve standard RL conventions by evolving, symbolically, algorithms represented as graphs, which leads to improved performance in simple tasks. However, none of those trained-from-scratch methods inherit correctness guarantees, limiting our certainty of the generality of their abilities. In contrast, our method, LPO, is meta-developed in a Mirror Learning space [18], where every algorithm is guaranteed convergence to an optimal policy. As a result to this construction, meta-training of LPO is easier than that of methods that learn "from scratch", and achieves great performance across environments. Furthermore, thanks to the clear meta-structure of Mirror Learning, LPO is interpretable, and lets us discover new learning strategies. This lets us introduce DPO—an efficient algorithm with a closed-form formulation that exploits the discovered learning concepts.

## 3 Background

In this section, we introduce the essential concepts required to comprehend our contribution—the RL and meta-RL problem formulations, as well as the Mirror Learning and Evolution Strategies frameworks for solving them.

### 3.1 Reinforcement Learning

**Formulation**   We formulate the reinforcement learning (RL) problem as a *Markov decision process* (MDP) [35] represented by a tuple $\langle \mathcal{S}, \mathcal{A}, R, P, \gamma, d \rangle$ which defines the experience of a learning agent as follows: at time step $t \in \mathbb{N}$, the agent is at state $\mathrm{s}_t \in \mathcal{S}$ (where $\mathrm{s}_0 \sim d$) and takes an action $\mathrm{a}_t \in \mathcal{A}$ according to its stochastic policy $\pi(\cdot|\mathrm{s}_t)$, which is a member of the policy space $\Pi$. The environment then emits the reward $R(\mathrm{s}_t, \mathrm{a}_t)$ and transits to the next state $\mathrm{s}_{t+1}$ drawn from the transition function, $\mathrm{s}_{t+1} \sim P(\cdot|\mathrm{s}_t, \mathrm{a}_t)$. The agent aims to maximise the expected value of the total discounted return,

$$\eta(\pi) \triangleq \mathbb{E}[\mathrm{R}^\gamma|\pi] = \mathbb{E}_{\mathrm{s}_0 \sim d, \mathrm{a}_{0:\infty} \sim \pi, \mathrm{s}_{1:\infty} \sim P}\Big[ \sum_{t=0}^{\infty} \gamma^t R(\mathrm{s}_t, \mathrm{a}_t) \Big]. \tag{1}$$

The agent guides its learning process with value functions that evaluate the expected return conditioned on states or state-action pairs

$$V_\pi(s) \triangleq \mathbb{E}[\mathrm{R}^\gamma|\pi, \mathrm{s}_0 = s] \quad \text{(the state value function)},$$

$$Q_\pi(s, a) \triangleq \mathbb{E}[\mathrm{R}^\gamma|\pi, \mathrm{s}_0 = s, \mathrm{a}_0 = a] \quad \text{(the state-action value function)}.$$

The function that the agent is concerned about most is the *advantage function*, which computes relative values of actions at different states,

$$A_\pi(s, a) \triangleq Q_\pi(s, a) - V_\pi(s). \tag{2}$$

**Policy Optimisation**   In fact, by updating its policy simply to maximise the advantage function at every state, the agent is guaranteed to improve its policy, $\eta(\pi_{\text{new}}) \geq \eta(\pi_{\text{old}})$ [35]. This fact, although requiring a maximisation operation that is intractable in large state-space settings tackled by deep RL (where the policy $\pi_\theta$ is parameterised by weights $\theta$ of a neural network), has inspired a range of algorithms that perform it approximately. For example, A2C [24] updates the policy by a step of policy gradient (PG) ascent

$$\theta_{k+1} = \theta_k + \frac{\alpha}{B} \sum_{b=1}^{B} A_{\pi_{\theta_k}}(s_b, a_b) \nabla_\theta \log \pi_{\theta_k}(a_b|s_b), \quad \alpha \in (0, 1), \tag{3}$$

estimated from a batch of $B$ transitions. Nevertheless, such simple adoptions of *generalized policy iteration* [35, GPI] suffer from large variance and instability [43, 33, 32]. Hence, methods that constrain (either explicitly or implicitly) the policy update size are preferred [31]. Among the most popular, as well as successful ones, is *Proximal Policy Optimization* [32, PPO], inspired by *trust region learning* [31], which updates its policy by maximising the PPO-clip objective,

$$\pi_{k+1} = \arg\max_{\pi \in \Pi} \mathbb{E}_{\mathrm{s} \sim \rho_{\pi_k}, \mathrm{a} \sim \pi_k}\Big[ \min\Big( \frac{\pi(\mathrm{a}|\mathrm{s})}{\pi_k(\mathrm{a}|\mathrm{s})} A_{\pi_k}(\mathrm{s}, \mathrm{a}), \text{clip}\Big( \frac{\pi(\mathrm{a}|\mathrm{s})}{\pi_k(\mathrm{a}|\mathrm{s})}, 1 \pm \epsilon \Big) A_{\pi_k}(\mathrm{s}, \mathrm{a}) \Big) \Big], \tag{4}$$

where the clip$(\cdot, 1 \pm \epsilon)$ operator clips (if necessary) the input so that it stays within $[1 - \epsilon, 1 + \epsilon]$ interval. In deep RL, the maximisation oracle in Equation (4) is approximated by a few steps of gradient ascent on policy parameters.

**Meta-RL** The above approaches to policy optimisation rely on human-possessed knowledge, and thus are limited by humans' understanding of the problem. The goal of *meta-RL* is to instead optimise the learning algorithm using machine learning. Formally, suppose that an RL algorithm $\text{alg}_\phi$, parameterised by $\phi$, trains an agent for $K$ iterations. Meta-RL aims to find the meta-parameter $\phi = \phi^*$ such that the expected return of the output policy, $\mathbb{E}[\eta(\pi_K)|\text{alg}_\phi]$, is maximised.

## 3.2 Mirror Learning

A Mirror Learning agent [18], in addition to value functions, has access to the following operators: the *drift function* $\mathfrak{D}_{\pi_k}(\pi|s)$ which, intuitively, evaluates the significance of change from policy $\pi_k$ to $\pi$ at state $s$; the *neighbourhood operator* $\mathcal{N}(\pi_k)$ which forms a region around the policy $\pi_k$; as well as sampling and drift distributions $\beta_{\pi_k}(s)$ and $\nu_{\pi_k}^\pi(s)$ over states. With these defined, a Mirror Learning algorithm updates an agent's policy by maximising the mirror objective

$$\pi_{k+1} = \underset{\pi \in \mathcal{N}(\pi_k)}{\arg\max} \, \mathbb{E}_{s \sim \beta_{\pi_k}} \left[ A_{\pi_k}(s, a) \right] - \mathbb{E}_{s \sim \nu_{\pi_k}^\pi} \left[ \mathfrak{D}_{\pi_k}(\pi|s) \right]. \tag{5}$$

If, for all policies $\pi$ and $\pi_k$, the drift function satisfies the following conditions:

1. It is non-negative everywhere and zero at identity $\mathfrak{D}_{\pi_k}(\pi|s) \geq \mathfrak{D}_{\pi_k}(\pi_k|s) = 0$,

2. Its gradient with respect to $\pi$ is zero at $\pi = \pi_k$,

then the Mirror Learning algorithm attains the monotonic improvement property, $\eta(\pi_{k+1}) \geq \eta(\pi_k)$, and converges to the optimal return, $\eta(\pi_k) \to \eta(\pi^*)$, as $k \to \infty$ [18]. A Mirror Learning agent can be implemented in practice by specifying functional forms of the drift function and neighbourhood operator, and parameterising the policy of the agent with a neural network, $\pi_\theta$. As such, the agent approximates the objective in Equation (5) by sample averages, and maximises it with an optimisation method, like gradient ascent. PPO is a valid instance of Mirror Learning, with the drift function:

$$\mathfrak{D}_{\pi_k}^{\text{PPO}}(\pi|s) \triangleq \mathbb{E}_{a \sim \pi_k} \left[ \text{ReLU}\left( \left[ \frac{\pi(a|s)}{\pi_k(a|s)} - \text{clip}\left( \frac{\pi(a|s)}{\pi_k(a|s)}, 1 \pm \epsilon \right) \right] A_{\pi_k}(s, a) \right) \right]. \tag{6}$$

While it is possible to explicitly constrain the neighbourhood of policy update [31], some algorithms do it implicitly. For example, as maximisation oracle of PPO (see Equation (4)) has a form of $N$ steps of gradient ascent with learning rate $\alpha$ and gradient clipping threshold $c$, it implicitly employs a neighbourhood of an Euclidean ball or radius $N\alpha c$ around $\theta_k$.

Different Mirror Learning algorithms can differ in multiple aspects such as sample complexity and wall-clock time efficiency [18]. Depending on the setting, different properties may be desirable. In this paper, we optimise for the return of the $K^{\text{th}}$ iterate, $\eta(\pi_K)$.

## 3.3 Evolution Strategies

Evolution Strategies [28, 29, ES] is a backpropagation-free approach to function optimisation. At their core lies the following identity, which holds for any continuously differentiable function $F$ of $\phi$, and any positive scalar $\sigma$

$$\nabla_\phi \mathbb{E}_{\epsilon \sim N(0,I)}[F(\phi + \sigma\epsilon)] = \frac{1}{\sigma} \mathbb{E}_{\epsilon \sim N(0,I)}[F(\phi + \sigma\epsilon)\epsilon], \tag{7}$$

where $N(0, I)$ denotes the standard multivariate normal distribution. By taking the limit $\sigma \to 0$, the gradient on the left-hand side recovers the gradient of $\nabla_\phi F(\phi)$. These facts inspire an approach of optimising $F$ with respect to $\phi$ without estimating gradients with backpropagation—for a random sample $\epsilon_1, \ldots, \epsilon_n \sim N(0,I)$, the vector $\frac{1}{n\sigma} \sum_{i=1}^{n} F(\phi + \sigma\epsilon_i)\epsilon_i$ is an unbiased gradient estimate. To reduce variance of this estimator, antithetic sampling is commonly used [26]. In the context of meta-RL, where $\phi$ is the meta-parameter of an RL algorithm $\text{alg}_\phi$, the role of $F(\phi)$ is played by the average return after the training, $F(\phi) = \mathbb{E}[\eta(\pi_K)|\phi]$. As oppose to the meta-gradient approaches described in Section 2, ES does not require backpropagation of the gradient through the whole training episode—a cumbersome procedure which, often approximated by the truncated backpropagation, introduces bias [38, 39, 25, 6, 23].

**(a)**

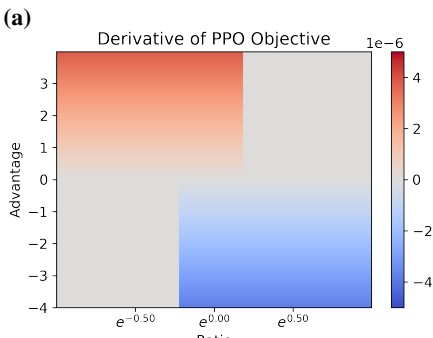

**(b)**

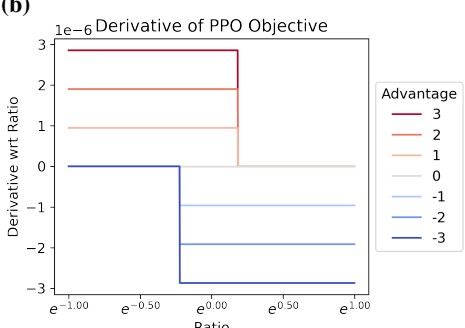

Figure 1: PPO objective visualisation: **(a)** is the heat map of the ratio derivative of the PPO objective, and **(b)** shows its slices for fixed advantage values. The algorithm encourages updates towards actions with positive values of the ratio derivative.

## 4 Methods

Our overall approach is to *meta-learn* a drift function to perform policy optimisation over a fixed episode length $K$. Hence, our meta-objective is the expected final return,

$$F(\phi) = \mathbb{E}[\eta(\pi_K)|\phi].$$

### 4.1 Drift Function Network

The drift function that we learn takes form $\mathfrak{D}_{\pi_k}(\pi|s) = \mathbb{E}_{a \sim \pi_k}[f_\phi(\mathbf{x})|s]$, where $f_\phi(\boldsymbol{x})$ is a fully-connected neural network parameterised by $\phi$. Our drift network is a function of the probability ratio between a candidate and the old policy, $r = \pi(a|s)/\pi_k(a|s)$, and of the advantage $A = A_{\pi_k}(s,a)$ (which we assume to be normalised across each batch). To ease learning complicated mappings, we include non-linear transformations of these arguments, ultimately forming the following input

$$\boldsymbol{x}_{r,A} = \left[(1-r),\ (1-r)^2,\ (1-r)A,\ (1-r)^2A,\ \log(r),\ \log(r)^2,\ \log(r)A,\ \log(r)^2A\right].$$

We perform ablations on different variants of the drift input in Appendix E. In general, we found that the performance of the learned drift function seems to be robust to different input parameterisations.

In order to guarantee that the neural network is a valid drift function, it must be the case that $f_\phi(\boldsymbol{x}_{r,A}) = 0$ and $\nabla_r f_\phi(\boldsymbol{x}_{r,A}) = \mathbf{0}$ whenever $r = 1$, and $f_\phi(\boldsymbol{x}_{r,A}) \geq 0$ everywhere. As in our model, $\boldsymbol{x}_{r,A} = 0$ whenever $r = 1$, the former condition is guaranteed by excluding bias terms from the network architecture. To meet the latter two conditions, we apply the ReLU activation at the last layer with a slight shift, $x \mapsto \text{ReLU}(x - \xi)$, where $\xi = 10^{-6}$.

To alleviate the difficulty of the meta-training, the main variant of our implementation initialises the drift function near the PPO one (see Section 5.2. We have found that this operation leads to a better performance and generalisation across different environments. Furthermore, such a setting directly tests for the optimality of the drift function of PPO. If PPO was optimal, the network could simply learn the zero mapping.

In principle, the drift function is not restricted to operate with probability ratios and advantages only; it could also accept other arguments, such as algorithm hyperparameters, statistics measuring the progress of training, or even task information. However, many of these vary greatly in the dimensionality or scale across different instances or types of environments. Furthermore, a large number of arguments would impede the analysis of the learnt drift function, and comparison to PPO, which are also goals of our work. Hence, we work with simple and transferable ratios and normalised advantage estimates. Nevertheless, for specific applications, it may be beneficial to consider other, possibly domain-specific, information.

### 4.2 Meta-Training the Drift Function Network

The meta-objective we optimise is the performance of the learner policy at the end of training:
$F(\phi) = \mathbb{E}\left[\eta(\pi_{\theta_K})|\text{alg}_\phi\right]$, where $\theta_K$ is the $K^{\text{th}}$ (last) iterate of the RL training under the Mirror

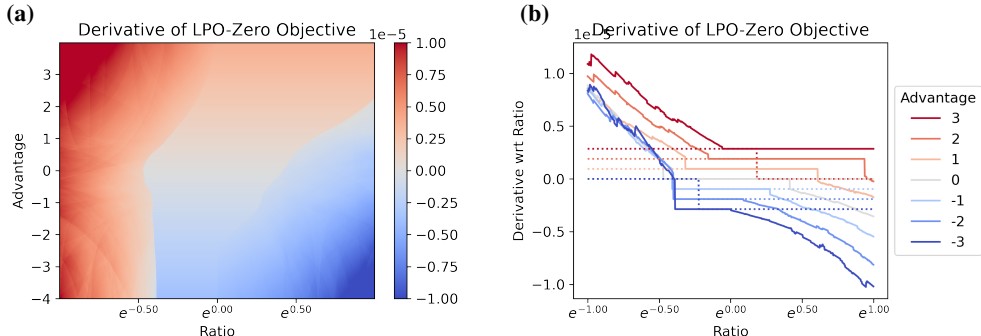

Figure 2: Visualisation of the LPO-Zero (learning from scratch) objective: **(a)** is the heat map of the ratio derivative of the LPO-Zero objective, and **(b)** shows its slices for fixed advantage values. The algorithm encourages updates towards actions with positive values of the ratio derivative.

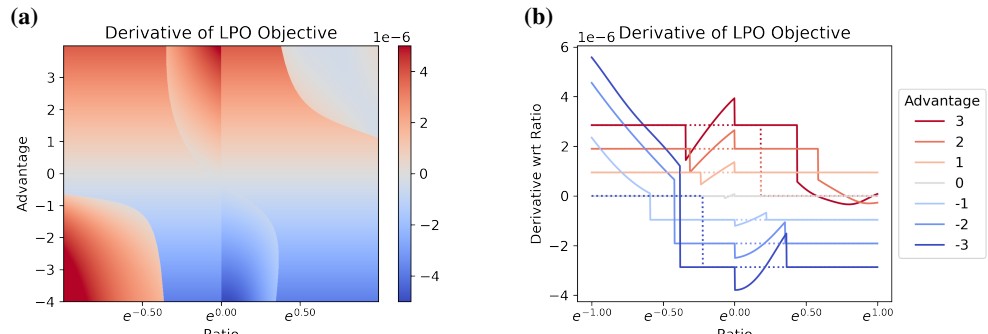

Figure 3: Visualisation of the LPO objective: **(a)** is the heat is the heat map of the ratio derivative of the LPO objective, and **(b)** shows its slices for fixed advantage values. The algorithm encourages updates towards actions with positive values of the ratio derivative.

Learning algorithm $\texttt{alg}_\phi$. The expectation is taken over the randomness of the initial parameter $\theta_0$ and stochasticity of the environment. We solve this problem using ES with antithetic sampling. At each generation (outer loop iteration), we sample a batch of perturbations of $\phi$, initialise the policy parameters $\theta_0$, and then train the policy under $\texttt{alg}_\phi$, using the drift function's parameter $\phi$, for $K$ iterations. At the end of the inner-loop training, we estimate the return of the final policy $\phi_{\theta_K}$, and use it to estimate the gradient of $\nabla_\phi F(\phi)$ as in Equation (7).

We meta-learn the drift function and evaluate policies trained by it in the *Brax* [8] physics simulator environments. Brax is designed to take advantage of parallel computation on accelerators, allowing us to roll out thousands of episodes in parallel, and train entire RL agents within minutes. Thus, it is well suited for this type of optimisation problem. Furthermore, we increase its efficiency by vectorising the *policy optimisation algorithm itself*, which lets us train hundreds of RL agents per minute using accelerators. We implement our method on top of the Brax version of PPO, which provides a Mirror Learning-friendly code template, keeping the policy architecture and training hyperparameters unchanged. For meta-training we use both *evosax* [19] and the *Learned_optimization* [22] libraries. For full details of meta-training see Appendix A.

## 5 Empirical Studies

We consider two different meta-training setups. First, as described in Subsection 5.1 we attempt to learn a drift function completely from scratch to investigate how similar it is to existing algorithms like PPO. Second, in Subsection 5.2 we ask whether we can learn a drift function that successfully generalises to multiple environments, if it is initialised near PPO.

## 5.1 Learning drift functions from scratch

In this setting, $f_\phi$ is a neural network with two hidden layers of size $256$ and a ReLU activation function. We meta-train it across $5$ Brax environments. We name the resulting algorithm LPO-Zero, and visualise it in Figure 2.

Interestingly, LPO-Zero appears to have learnt a few PPO-like features, as can be observed on Figure 2. For example, it appears to have learnt to clip the update incentive at a specific ratio threshold, much like PPO; however, it only does so for negative advantages. Nevertheless, LPO-Zero largely underperforms with respect to LPO, and possibly requires much more training to catch up. We present figures with performance evaluation of LPO-Zero in Appendix B.

## 5.2 Learning with the PPO Initialisation

In this setting, $f_\phi$ is a small neural network, with a single hidden layer with $128$ neurons, with bias terms removed, and a $\mathtt{tanh}$ activation function. Here, we add PPO to the output of the last hidden layer, before passing it to the shifted ReLU,

$$f_\phi(\boldsymbol{x}_{r,A}) = \mathrm{ReLU}\Big(\tilde{f}_\phi(\boldsymbol{x}_{r,A}) - \xi + \mathrm{ReLU}\left((r - \mathrm{clip}(r, 1 \pm \epsilon)) \cdot A\right)\Big), \tag{8}$$

where $\tilde{f}_\phi(\boldsymbol{x}_{r,A})$ is the output of the last hidden layer of the drift network. As such, the resulting drift function similar to that of PPO at initialisation.

Surprisingly, we have found that meta-training in a **single** environment is sufficient to generate drift functions whose abilities transfer to unseen tasks. Moreover, we found that the learnt drifts generally display similar characteristics. For readability, we chose the drift function that was trained on Ant, whose induced algorithm we refer to as *Learnt Policy Optimisation* (LPO). Visualisations and results for drift functions trained on other environments can be found in the Appendix C.

The results on Figure 6 show that LPO, trained only on Ant, outperforms PPO in unseen environments. Furthermore, the Brax PPO implementation uses different hyperparameters, such as the number of update epochs and the total number of timesteps, for each of the tasks. This means that LPO, which was trained on *Ant* with hyperparameters associated to it, is robust not only against new environments, but also against new hyperparameters. We visualise the derivative of the LPO loss in Figure 3, which enables us to derive an analytical version of it in Section 6.

## 6 Analysis of LPO

In this section, we analyse the two key features that are consistently learnt and contribute most to LPO's performance. We then interpret their effect on *policy entropy*, and the *update asymmetry* discovered by LPO, through which it differs largely from PPO (recall Figure 3 for visualisation).

**Rollback for negative advantage.** In the bottom-left quadrant of the heat map, which corresponds to $A < 0$ and $r < 1$ (negative advantage and decrease in action probability), we observe that the ratio derivative of the LPO objective is positive in a large region, roughly corresponding to $r < 1 - \epsilon$. This implies that actions which fall into this quadrant, although seemingly not appealing, are encouraged to be taken by the agent, which can be interpreted as a form of *rollback* [37]. Hence, LPO learns to decrease $r$ down to $1 - \epsilon$, but unlike PPO, encourages $r$ to stay precisely around that value. By doing so, LPO prevents the agent from giving up on actions that appear poor at the moment, and encourages it to keep exploring them at a moderate frequency.

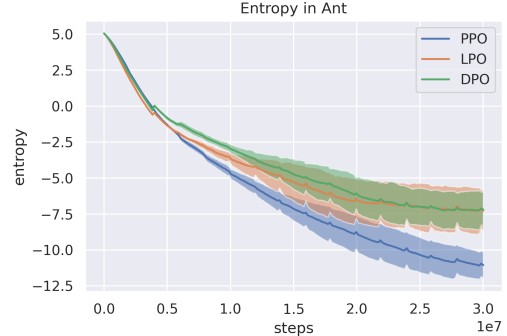

Figure 4: Entropy comparison, throughout training on Ant, between PPO (blue), LPO (orange), and DPO (green, see Section 7) across 10 seeds. Error bars denote standard error. While the entropy of all methods decrease throughout training, the entropy of policy learned by both LPO and DPO remain significantly higher than that of PPO.

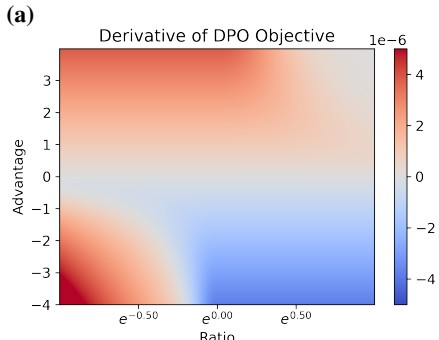
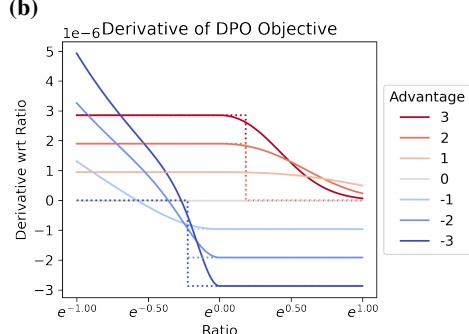

Figure 5: Visualisation of the DPO objective: **(a)** is the heat is the heat map of the ratio derivative of the DPO objective, and **(b)** shows its slices for fixed advantage values. Positive values of the derivative encourage updates towards the action.

**Cautious optimism for positive advantage.**    The upper-right quadrant corresponds to $A > 0$ and $r > 1$ (positive advantage and increase in action probability), which is induced by actions that seem the most appealing to update to. Nevertheless, LPO is cautious in doing so, gradually decreasing the pace of its update towards them, and eventually abstaining from chasing the most extreme advantage values—these may come from critic errors. We would like to highlight that this view of LPO on positive advantages in much more sophisticated than that of PPO, which simply removes any incentive from updating towards actions with $r > 1 + \epsilon$, and thus can be viewed as *optimistic* relative to PPO.

**Implicit entropy maximisation.**    Together, these two central features of LPO encourage the agent to spread its policy probability mass moderately over all actions, thus leading to larger entropy and allowing for richer exploration. Thus, LPO ***has implicitly discovered entropy maximisation***, which we demonstrate in Figure 4. We would like to highlight that LPO achieved this with its drift function only, without artificially augmenting the original RL objective with an entropy bonus [11, 12]

**Update asymmetry.**    LPO learns *asymmetric* features that respect a natural *asymmetry* of behaviour change in RL: increasing $r$ for positive advantage $A$ may encourage exploration of a newly-found action *or* strengthen a dominant action, whereas decreasing $r$ for negative $A$ will always discourage exploration of that action *and* strengthen a dominant action. In this context, the two discussed features of LPO make it completely unlike PPO, which clips the update incentives symmetrically around the origin.

**Secondary Features.**    LPO, but not LPO-Zero, appears to consistently learn objectives with gradient spikes around $r = 1$ in the upper left and lower right quadrants. Nevertheless, adding them to our analytic model of LPO did not improve performance. We speculate, therefore, that these spikes are mostly artifacts of the network parameterisation.

# 7    Discovered Policy Optimisation: A New RL Algorithm Inspired By LPO

In this section, building upon concepts that LPO has discovered, we introduce a novel algorithm—Discovered Policy Optimization (DPO).

## 7.1    The Discovered Drift Function Model

Combining the key features identified in Section 6, we construct a closed-form model of LPO that can easily be implemented with just a few lines of code on top of an existing PPO implementation. We name the new algorithm *Discovered Policy Optimisation* (DPO) because we have not derived it—it was instead discovered in the meta-learning process. DPO is a Mirror Learning algorithm, with a drift function that takes different functional forms, depending on the sign of advantage $A$, as dictated by the update asymmetry principle from the previous section. Specifically, we have found

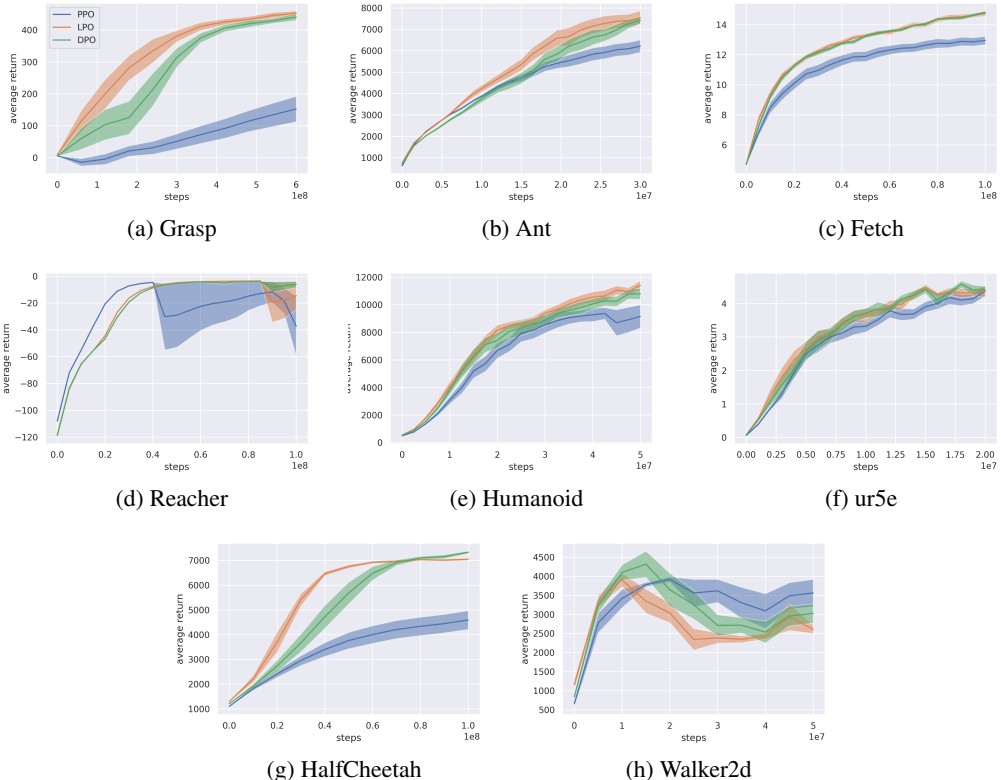

Figure 6: Performance comparison between PPO (blue), LPO (orange), and DPO (green) in Brax environments. The curves represent mean evaluation return across 10 random seeds, with error bars showing standard error of the mean. Both LPO and DPO beat PPO across most environments even though they were only meta-trained on Ant, and make use of no hyper parameter tuning. Furthermore, both LPO and DPO are more consistent across runs.

that the (parameter-free) drift function

$$f(r, A) = \begin{cases} \text{ReLU}\big((r-1)A - \alpha \tanh((r-1)A/\alpha)\big) & A \geq 0 \\ \text{ReLU}\big(\log(r)A - \beta \tanh(\log(r)A/\beta)\big) & A < 0 \end{cases} \tag{9}$$

faithfully reproduces the key features of LPO (cautious optimism and rollback) for appropriate constants $\alpha = 2, \beta = 0.6$ (see Appendix D for verification of the drift conditions). We visualise DPO in Figure 5 and note that even the "crossing-over" of gradient slices of LPO on Figure 3 is faithfully reproduced.

## 7.2 Results

We compare DPO to PPO and LPO on a variety of Brax environments in Figure 6. We use the PPO implementation provided by Brax, with an addition of advantage normalisation as we observed it to improve the method's performance across the majority of the environments. Our methods also use this implementation technique. We further demonstrate on out-of-distribution performance by evaluating performance on the Minatar environments [20, 41] in Figure 7.

While evaluating DPO, similarly to LPO, we do not re-tune *any* hyperparameters that were originally selected for *PPO* in Brax. The results on Figure 6 show that DPO matches the performance of LPO and outperforms PPO on the evaluated environments, *despite being a two-line analytic model of LPO based on two key features*. This enables RL practitioners to implement DPO as easily as PPO with a performance on par with our best learnt drift function.

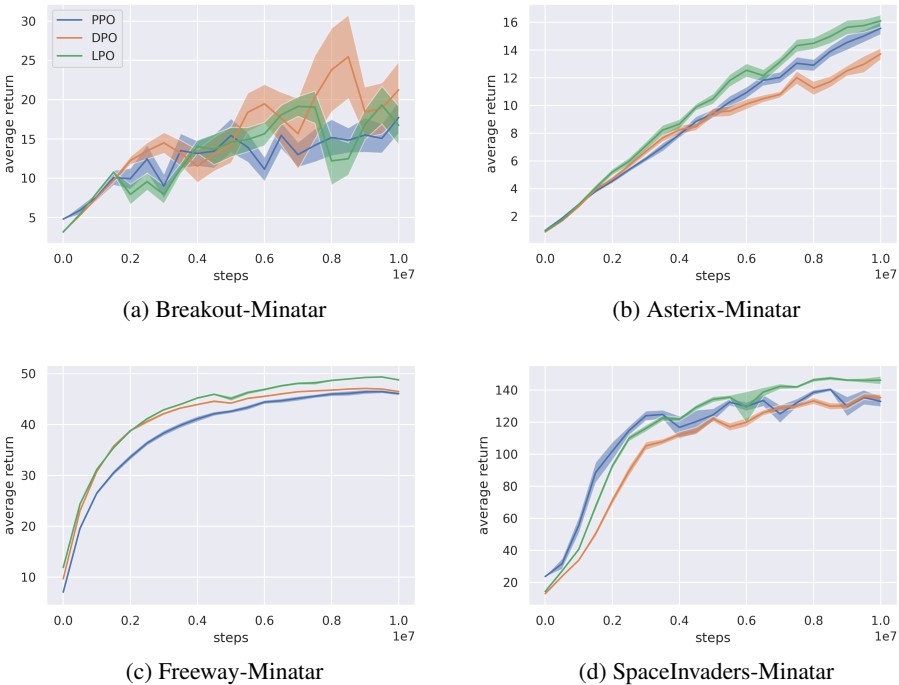

(a) Breakout-Minatar

(b) Asterix-Minatar

(c) Freeway-Minatar

(d) SpaceInvaders-Minatar

Figure 7: Performance of PPO, DPO, and LPO on the Minatar environments [20, 41]. We tuned the hyperparameters for PPO and re-used those tuned hyperparameters for DPO and LPO. The curves represent mean evaluation return across 10 random seeds, with error bars showing standard error of the mean.

## 8   Conclusion

In this paper, we approached the problem of *algorithm discovery* by restricting our meta-learning to the space of valid *Mirror Learning* algorithms. Specifically, we optimised a drift function parameterised by a neural network, which we trained with *Evolution Strategies*. We consider this work to be an example of the new, promising paradigm of RL algorithm discovery. Namely, our strategy was to develop a high-performing RL algorithm by combining theoretical insights with large-scale computational techniques. As a result of the training, we obtained a theoretically sound method that we named *Learnt Policy Optimisation* (LPO), which outperforms a state-of-the-art baseline (PPO) in unseen environments, and with unseen hyperparameter settings. After analysing the learned features discovered by LPO, we introduced *Discovered Policy Optimisation* (DPO)—a closed-form approximation to LPO. Our experimental results show that DPO matches LPO in performance and robustness to hyperparameter settings. However, a possible weakness of this approach is that it could overfit to specific code-level details of the implementation. For example, LPO is built on Brax's PPO implementation, which makes numerous design decisions to optimise for wall-clock time rather than sample efficiency. In the future, we plan to expand the variety of inputs to the learnt drift function, as well as to parameterise and meta-learn other attributes of Mirror Learning. We expect these advancements to provide more insights into policy optimisation, ultimately resulting in more robust and better performing RL algorithms.

## Acknowledgments and Disclosure of Funding

Christian Schroeder de Witt is sponsored by the Cooperative AI Foundation. Compute for this project was partially run on Oxford's Advanced Research Cluster (ARC). This work used the Cirrus UK National Tier-2 HPC Service at EPCC funded by the University of Edinburgh and EPSRC (EP/P020267/1). This work was also supported by an Oracle for Research Cloud Grant (19158657).

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
